

# What are the Andean Colombian anurans? Empirical regionalization proposals *vs.* observed patterns of compositional dissimilarity

Jorge Mario Herrera-Lopera[1,2], Viviana Andrea Ramírez Castaño[2,3] and Carlos A. Cultid-Medina[4,5]

[1] Laboratório de Herpetologia Tropical, Programa de Pós-Graduação em Ecologia e Conservação da Biodiversidade, Universidade Estadual de Santa Cruz, Ilhéus, Bahia, Brasil
[2] Grupo de Investigación en Biodiversidad y Recursos Naturales (BIONAT), Semillero de investigación en Biodiversidad y Conservación de Paisajes Urbanos (OIKOS), Universidad de Caldas, Manizales, Caldas, Colombia
[3] Departamento de Ciencias Biológicas, Universidad de Caldas, Manizales, Caldas, Colombia
[4] CONACYT, Mexico City, Mexico City, Mexico
[5] Red de Diversidad Biológica del Occidente Mexicano, Instituto de Ecología, A.C (INECOL) Centro Regional del Bajío, Pátzcuaro, Michoacán, México

Corresponding author
Carlos A. Cultid-Medina, carlos.cultid@inecol.mx

## ABSTRACT

**Background:** Defining Andean anurans through their altitudinal limits has been a common practice in species lists, studies of responses to climate change among others, especially in the northern Andes. At least three proposals to differentiate Andean anurans from lowland anurans through elevation and at least one to differentiate Andean anurans from high mountain anurans have been formulated. However, the most frequently used altitudinal limits are not based on theoretical or numerical support, but on observations or practical definitions. Additionally, these proposals have been applied equally to different portions of the Andes, ignoring the fact that even between slopes of the same mountain, environmental conditions (and therefore the distribution of species) may differ. The objective of this work was to evaluate the concordance between the altitudinal distribution of anurans in the Colombian Andes and four different altitudinal delimitation proposals.
**Methods:** We constructed our study area in a manner that allowed us to include species from the Andean region (as traditionally defined) and adjacent lowlands, because if the boundaries criteria were applied they would separate the species of the latter by themselves. We divided the study area into eight entities according to the watershed and the course of the most important rivers. We conducted a bibliographic search for all anurans in the cordilleras and inter-Andean valleys of Colombia and complemented the search with information on anurans for the region available in the GBIF. After curing the species distribution points, we generated elevation bands of 200 m amplitude for both the study area and for each Andean entity. Subsequently, we performed a cluster analysis to evaluate the grouping of the elevation bands according to their species composition.
**Results:** In none of the cases (neither for the entire study area nor for any of the entities) we found a correspondence of any of the traditionally used boundaries and the altitudinal distribution of Anurans in the Andean region of Colombia. Instead, on

average, the altitudinal delimitation proposals arbitrarily spanned the altitudinal distribution of about one third of the species distributed in the study area.

**Conclusions:** We suggest that, although, based on our results, some Andean entities can be divided according to the altitudinal composition of the species that occur in them, we did not find any results that support the idea of a generalizable altitudinal limit for the Colombian Andes. Thus, to avoid biases in studies that may later be used by decision makers, the selection of anuran species in studies in the Colombian Andes should be based on biogeographic, phylogenetic or natural history criteria and not on altitudinal limits as they have been used.

## INTRODUCTION

The Colombian Andes belong to the northern Andes region, which extends from the Amotape-Huancabamba depression between Peru, and Ecuador (5 °S), to the Caribbean plate contact point, in the Mérida mountain range in Venezuela (*ca.* 12 °N) (*Graham, 2009*). The Colombian Andes comprise *ca.* 66% of the northern Andes and, given the topographic complexity, represent one of the most diverse regions in the world. In particular, the northern Andes host 27.6% of the South American anuran species, with 73% of this richness concentrated in Colombia, ranking the country as the second richest in anuran species worldwide (*Cochran & Goin, 1970*; *Armesto & Señaris, 2017*; *Frost, 2022*; *Acosta Galvis, 2019*). Despite the high species richness of the region, the definition of which should be considered an Andean anuran is unclear.

Historically, at least four empirical definitions have been postulated to circumscribe Andean anurans. Three have defined minimum elevation limits to differentiate Andean from lowland anurans, and a fourth has been postulated to differentiate Andean anurans from those that are exclusive to high mountain ecosystems (high mountain anurans). All definitions have been used to delimit Andean anurans in regional species lists, studies of response to climate change scenarios and local species lists, but their arbitrary use may have repercussions for decision-making and knowledge of the fauna of the region (*Péfaur & Rivero, 2000*; *Bernal & Lynch, 2008*; *Armesto & Señaris, 2017*). Although, in practical terms, trying to define complex communities within arbitrary geographic boundaries brings us back to Clements and Gleason's discussion of whether communities are open or closed systems (see *Begon, Townsend & Harper, 2006*), the truth is that in practical terms the use of arbitrary delimitation for a group, (understanding its advantages and limitations), should ease to obtain general responses to specific scenarios.

First, *Duellman (1979)* considered Andean anurans to be distributed above 1,000 m asl in elevation, excluding the species mainly distributed in lowlands and with only peripheral occurrence above this elevation. This delimitation was formulated for the single purpose of compiling the first list of anurans of South America. Although this proposal is not justified in any formal theory or observation, later authors accepted and used it as an approach for delimiting Andean anurans (*e.g.*, *Bernal & Lynch, 2008*; *Armesto & Señaris, 2017*). On the

other hand, based on the distribution of the former genus *Eleutherodactylus* (whose species are currently assigned to different genera within the families Craugastoridae, Eleutherodactylidae and Strabomantidae) in the Colombian Andes, *Lynch (1999)* proposed a vertical classification, where anurans were considered as Andean if they were distributed above 900 m asl. Although this classification was proposed considering a portion of the anuran diversity of the northern Andes, Lynch suggested that the proposal could also apply to other anuran groups. Some authors have therefore used the delimitation to differentiate Andean anurans based on elevation (*e.g.*, *Armesto & Señaris, 2017*).

Meanwhile, and almost simultaneously with the proposal by *Lynch (1999)*, *Péfaur & Rivero (2000)* analyzed the spatial distribution of the anurans of Venezuela, and defined the elevation limit between lowland and Andean anurans at 500 m asl. Again, although this proposal was not justified in any formal theory or observations, some authors have used it as a criterion for differentiating Andean anurans (*e.g.*, *Anderson et al., 2011*; *Meza-Joya & Torres, 2016*).

In the case of Colombia, the application of any of the three above proposals to differentiate Andean anurans from lowland anurans presented above (also called lower boundary proposals), would imply the exclusion of anurans distributed in portions of the inter-Andean valleys (*i.e.*, Cauca River valley and Magdalena River valley).

In addition, authors such as *Lynch (1999)* and *Navas (2002)* have proposed an upper elevation limit for Andean anurans. These proposals suggest the elevation limit between Andean anurans and high mountain anurans (= High Andean anurans) at nearly 3,000 m asl. The approaches by *Lynch (1999)* and *Navas (2002)* are linked to the idea that mountain peaks "behave as islands" and are based on the distribution of the ancient genus *Eleutherodactylus* in the western Colombian mountain range (*Lynch, 1999*), and on ecophysiological observations on species of anurans in the Andes (*Navas, 2002*). This delimitation has been used in the subsequent literature to delimit high mountain species and conduct vulnerability studies in the face of climate change scenarios (*e.g.*, *Acosta-Galvis, 2015*; *Agudelo-Hz, Urbina-Cardona & Armenteras-Pascual, 2019*). Despite the apparent differences between the four delimitation proposals mentioned above, there is still no consensus as to which proposal(s) is/are most appropriate, and all of them continue to be used interchangeably and extensively to delimit Andean anurans, when it is in fact unclear whether it is appropriate to use such a generalized definition of Andean anurans.

Compositional dissimilarity (beta diversity) has been used as a tool for the delimitation of biogeographic units (*e.g.*, *Hernández-Camacho, 1992*; *Lynch, Ruiz-Carranza & Ardila-Robayo, 1997*; *Lynch, 1999*; *Morrone, 2014*; *Rahbek et al., 2019a*), which are frequently used to explain the patterns of distribution and diversification of species on a large scale, as well as to delimit priority conservation areas and regions of radiation and endemism (*e.g.*, *Whitehead, Bowman & Tideman, 1992*; *Whiting et al., 2000*; *Chen & Bi, 2007*; *Rahbek et al., 2019b*). In amphibians and reptiles, compositional dissimilarity has been used to delimit biogeographic regions, model endemism zones and test the consistency of biogeographic proposals (*Lynch, Ruiz-Carranza & Ardila-Robayo, 1997*; *Nori, Díaz Gómez & Leynaud, 2011*; *Vasconcelos et al., 2019*). In this sense, amphibians have been used as model organisms in climate change scenarios and biogeographic regionalizations (*Chen &*

*Bi, 2007*; *Acosta-Galvis, 2015*; *Agudelo-Hz, Urbina-Cardona & Armenteras-Pascual, 2019*). To our knowledge, there are no studies to date which have tested the different proposals for delimiting the altitudinal distribution of anurans in the Andes (see above), although choosing one or another proposal arbitrarily can potentially lead to different results and conclusions. Likewise, we are not aware of studies with any other vertebrate groups in the Andes, where this type of altitudinal boundaries are tested numerically.

Many authors have found variations in the elevations at which different ecosystems occur within the same mountain system slopes (*Rahbek et al., 2019a*, *2019b*). This phenomenon is explained by changes in physical variables (*i.e.* wind, humidity, cloudiness) among different slopes, causing units of the same mountain system to behave differently and, consequently, to differ in terms of the associated biota on each slope (*Narváez-Bravo & León-Aristízabal, 2001*; *Kattan et al., 2004*; *Rahbek et al., 2019b*). These differences are enough to consider each range as a biologically independent sample within the same topographic region (*Kattan & Franco, 2004*). The northern portion of the Andes in Colombia is considered the most bioclimatically and topographically complex of the Andean system (*Kattan et al., 2004*). However, the proposals for delimiting Andean anurans do not consider these variations between the entities that compose the Andes (*i.e.*, each of the mountain ranges and the inter-Andean valleys) and have not evaluated whether the proposals are generalizable or whether their application varies depending on the mountain slope or Andean component to which they are to be applied.

Our aim was to evaluate the consistency of the proposed altitudinal delimitations of Péfaur & Rivero: 500 m asl, Lynch: 900 m asl, Duellman: 1,000 m asl and Navas—Lynch: 3,000 m asl, with the distribution of anurans in the Colombian Andes. Assuming that the above altitudinal limits actually reflect the altitudinal distribution of anurans in Colombia and considering that these delimitation proposals do not differ substantially in their lower limits (*i.e.*, 500, 900 and 1,000 m asl), we would expect to obtain three different groups of anurans along elevation gradients across the Andes of Colombia: (1) lowland anuran species (approximately between 500 and 1,000 m asl), (2) Andean anuran species (at least between 1,000 m to 3,000 m asl) and (3) high Andean anuran species (>3,000 m asl). In addition, the conformation of these elevation groups should be consistent or similar across different Andean slopes in Colombia (*i.e.*, applicable to all of the Colombian Andes).

## MATERIALS AND METHODS

### Study area

The Colombian Andes are divided into three mountain ranges: the western, central, and eastern ranges, which diverge from a high-rising massif (Macizo Colombiano) in the Colombian southwest (*Irving, 1975*; *Kattan et al., 2004*). The western range (Cordillera Occidental) is separated from the Pacific Ocean by a narrow strip of rainforest (*Kattan et al., 2004*). The western and central (Cordillera Central) ranges are separated by the Cauca River valley with an approximate elevation of 1,000 m asl in its middle zone and elevation decreases in both northern and southern valley zones to about 200 m asl (*Kattan et al., 2004*). The central and eastern (Cordillera Oriental) ranges are separated by the

Magdalena River valley, which has an elevation in its middle zone of 500 m asl, decreasing in the north of the valley to approximately 80 m asl (*Duellman, 1979*; *Hernández-Camacho, 1992*; *Kattan & Franco, 2004*), the eastern slope of the eastern range is connected to the Orinoco and Amazon regions. Considering the Colombian Andes (as defined above) and its surrounding area, a study area was defined for the purposes of this work. Thus, the Chocó-Magdalena and Norandina provinces (*Hernández-Camacho, 1992*) were considered and an extension to the east of the Cordillera Oriental was added, including elevations equal to or higher than 200 m asl. This area has the advantage of including the inter-Andean valleys, the Colombian Pacific corridor and the foothills of the Cordillera Oriental towards Orinoco and Amazonia, plus the northern limit is indicated by the boundaries of the Chocó-Magdalena and Pre-Caribbean Arid Belt provinces of *Hernández-Camacho (1992)* (Fig. 1, see data availability section).

Since our aim was to evaluate the consistency of the altitudinal boundaries and anuran distribution in the Colombian Andes at the level of the entire study region and at the level of each of the slopes of the mountain ranges, we divided the study area into eight parts (called **entities** throughout this work). These entities were delimited according to the watershed (line of highest elevation) of the mountain ranges and the Macizo Colombiano and the course of the main rivers in the Colombian Andean region (*i.e.*, Magdalena, Cauca, Patía and Caquetá Rivers). These entities were: Macizo Colombiano: western slope (MCW) and eastern slope (MCE); Cordillera Occidental: western slope (COCW) and eastern slope (COCE); Cordillera Central: western slope (CCW) and eastern slope (CCE); and Cordillera Oriental: western slope (CORW) and eastern slope (CORE) (Fig. 1).

## Anuran records from the Colombian Andes

Records of anuran distribution in the Colombian Andes were obtained in two different approaches:

(i) Literature search: Anuran species were compiled from the following sources: (i) original descriptions of the species, (ii) articles published in scientific journals specifying museum codes and collection sites, (iii) short notes published in scientific journals in which the distribution range of some species is extended with the support of a museum number and, (iv) databases from the Instituto de Ciencias Naturales of the Universidad Nacional de Colombia (*ICN, 2004*), Instituto Alexander von Humboldt (*IAvH, 2018*), Universidad del Valle (*Herpetology Collection of the Universidad del Valle, 2016*) and the Museo de Historia Natural de la Universidad de Caldas MNH-UCa (*Serna-Botero & Ramírez-Castaño, 2017*). For records in which the coordinates were imprecise or not available, were used an approximation to the nearest town (municipality, township or village) using Google Earth Pro (*Google Earth, 2018*).

(ii) GBIF: Anuran distribution data were downloaded using the *rgbif* package of the R language and environment (*R Core Team, 2022*; *Chamberlain et al., 2023*). Considering only records for Colombia, with coordinates and from preserved specimens (see search DOI in *GBIF, 2022*).

The records were curated using the functions provided by the *scrubr* and *CoordinateCleaner* packages of the R language and environment (*Zizka et al., 2019*;
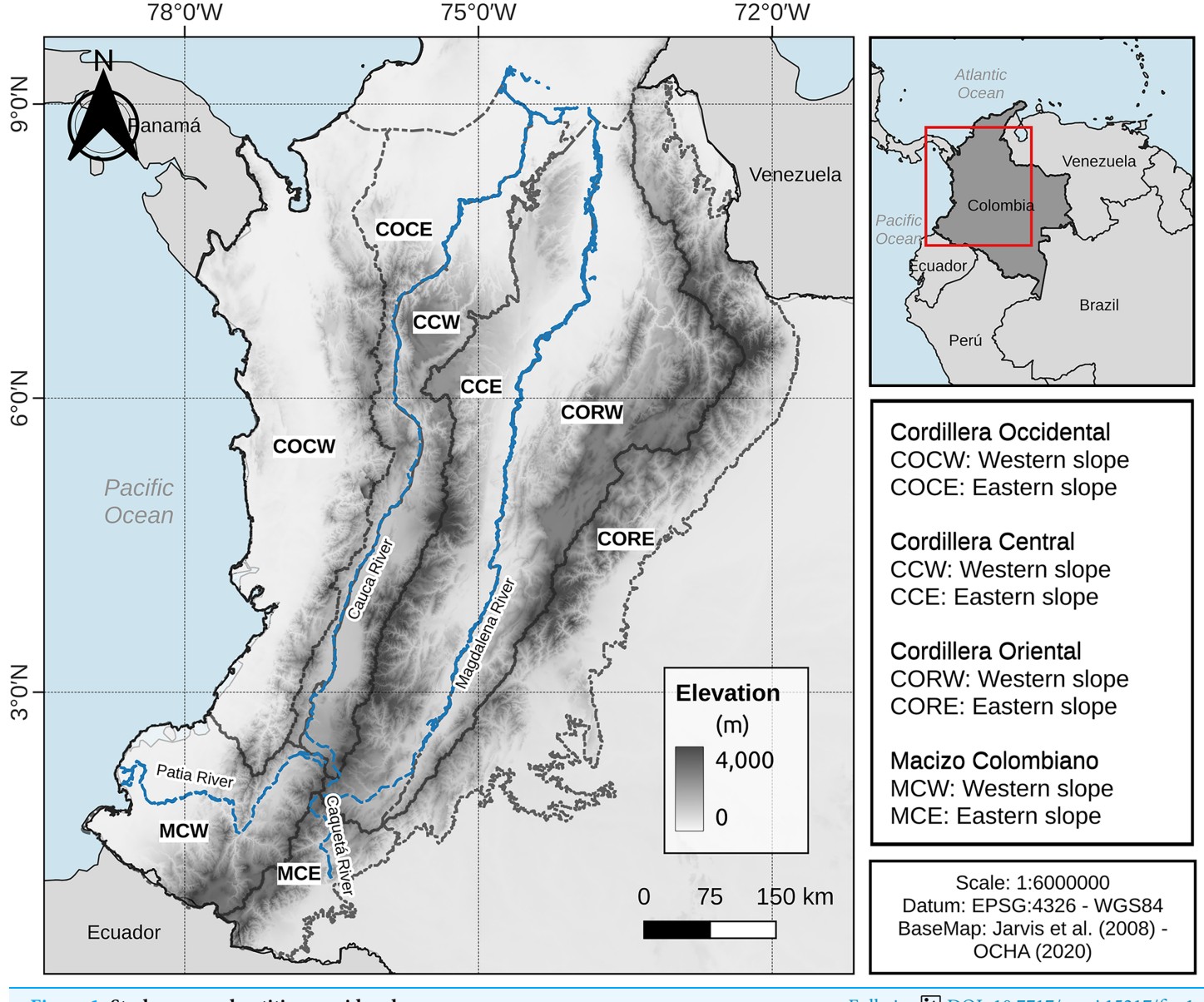

**Figure 1 Study area and entities considered.**

*Chamberlain, 2022*; *R Core Team, 2022*), and duplicate records were removed to preserve only those localities unique to each species. Taxonomy to family, genus and species level was corrected using the American Museum database and the *AmphiNom* package of the R language and environment (*Liedtke, 2019*; *Frost, 2022*; *R Core Team, 2022*, see Data Availability section).

We assigned the elevation data to each species record using a digital elevation model (DEM, resolution ~90 m) obtained from Consortium for Spatial Information database (*Jarvis et al., 2008*) and Qgis (*QGISC, 2022*). All taxonomic nomenclature and species are updated to December 2022.

## Data analyses

To evaluate the consistency of the four altitudinal delimitation proposals (*i.e.*, *Duellman, 1979*; *Lynch, 1999*; *Péfaur & Rivero, 2000*; *Navas, 2002*) and distribution of anuran species in the Colombian Andes, we analyzed compositional dissimilarity patterns throughout the study area and the above-defined entities (see Study Area section). For this, the species were grouped in elevation bands of 200 m each. Using these bands we constructed clusters for the study region and for each of the entities using the UPGMA method (*Legendre & Legendre, 2012*; *Suzuki, Terada & Shimodaira, 2019*) and the Jaccard dissimilarity index (*Carvalho et al., 2013*). Since our aim was to examine the elevation bands grouping, we used only the total beta ($\beta_{cc}$) measure (based on the Jaccard dissimilarity index) from *Carvalho et al. (2013)* proposal, which can be understood as the sum of beta diversity explained by species turnover and beta diversity explained by the difference in species richness. Total beta measure was calculated using the *BAT* package of the R language and environment (*Cardoso et al., 2022*; *R Core Team, 2022*).

The grouping of each cluster was evaluated using the statistical GAP method (nboot = 10,000), through the *NbClust* package of the R language and environment (*Tibshirani, Walther & Hastie, 2001*; *Charrad et al., 2014*; *R Core Team, 2022*). Support for each cluster group was evaluated using Jaccard bootstrap ($J_b$, nboot = 10,000) (*Legendre & Legendre, 2012*), using the R language and environment and the package *pvclust* (*Suzuki, Terada & Shimodaira, 2019*; *R Core Team, 2021*).

To evaluate the percentage of species, genera and families of anurans in the study area and in each entity, whose altitudinal distribution was spanned by each one of the proposed delimitations (that is, whose distribution has minimum values below some delimitation and maximum above said delimitation), the altitudinal distribution ranges were extracted for each species and the number of species (or genera or families) spanned by the proposal and divided by the total number of species (genera or families) in the study area or entity. To compare the altitudinal distribution of the anuran families in the study area with the clustering of the elevation bands observed in the cluster and the expected clustering under the delimitation of the biogeographic provinces, we constructed heat-maps for the anuran families *vs.* the elevation bands for each of the entities in which the cluster analysis detected clusters. The plot to study area and each entities was constructed using the *ggplot2* package of the R language and environment (*Wickham, 2016*; *R Core Team, 2022*).

## RESULTS

A total of 107,180 distribution records were obtained (literature search: 34,388 + GBIF: 72,792), corresponding to 17,653 unique localities for 675 species, 76 genera and 14 families after record cleaning (Table 1). This study thus covered 81.33%, 88.37%, and 93.33% of the species, genera and families, respectively, of native anurans known from Colombia (*Frost, 2022*).

The Andean entity with the highest species richness was the western slope of the Cordillera Occidental (COCW, 286 spp.), followed by the eastern slope of the Cordillera Central (CCE, 235 spp.) and the eastern slope of the Cordillera Oriental (CORE, 235 spp.) (Table 1).

**Table 1 Total number of species, genera and families distributed in the study area and included in the analysis.**

| Entity | Species richness | Genera richness | Family richness |
|--------|------------------|-----------------|-----------------|
| SA | 675 | 76 | 14 |
| COCW | 286 | 49 | 14 |
| COCE | 182 | 45 | 13 |
| CCW | 210 | 48 | 12 |
| CCE | 235 | 48 | 13 |
| CORW | 173 | 45 | 13 |
| CORE | 235 | 53 | 12 |
| MCW | 111 | 34 | 12 |
| MCE | 133 | 39 | 10 |

**Note:**
SA, Study area; COCW, Cordillera Occidental western slope; COCE, Cordillera Occidental eastern slope; CCW, Cordillera Central western slope; CCE, Cordillera Central eastern slope; CORW, Cordillera Oriental western slope; CORE, Cordillera Oriental eastern slope; MCW, Macizo Colombiano western slope; MCE, Macizo Colombiano eastern slope.

The cluster analysis for the study area detected three different groups. The first one was formed by the elevation bands between 200–2,400 m asl ($J_b = 0.89$), the second one was formed by the bands between 2,600–3,800 m asl ($J_b = 0.81$) and the third one grouped the bands between 4,000– 4,400 m asl ($J_b = 0.90$) (Fig. 2). The detailed clusters for the study area and each of its entities can be seen in Fig. S1.

For the Cordillera Occidental, COCW was formed by three elevation groups. The first one was formed by the bands between 200–3,600 m asl and the 4,000 m asl band ($J_b = 0.87$), the second one was formed by the 3,800 m asl band ($J_b = 0.64$) and the third one by the 4,200 m asl band ($J_b = 0.64$). On the other hand, COCE formed a single group (no groups were detected) according to the statistical GAP method ($J_b = 1$) (Fig. 2).

In the Cordillera Central, CCW had three groups, the first one formed by the bands of 200, 600, and 1,000–1,800 m asl ($J_b = 0.70$), the second one formed by the bands of 400 and 800 m asl ($J_b = 0.71$) and the third one, formed by the bands between 2,000–4,000 m asl ($J_b = 0.70$). Meanwhile, CCE showed three groupings, the first formed by the elevation bands between 200–2,200 m asl ($J_b = 0.91$), the second one was formed by the bands between 2,400–4,200 m asl ($J_b = 0.84$) and the third one by the 4,400 m asl band ($J_b = 0.52$) (Fig. 2).

Regarding the Cordillera Oriental clusters, CORW presented three groups, the first one formed by the bands between 200–2,200 m asl ($J_b = 0.79$), the second one was formed by the elevation bands between 2,400–3,800 m asl ($J_b = 0.86$) and the third one, formed by the bands from 4,000–4,200 m asl ($J_b = 0.83$). Also, CORE presented three groups, the first one formed by the bands between 400–2,400 m asl ($J_b = 0.97$), the second one was formed by the elevation bands between 2,600–3,800 m asl ($J_b = 0.86$) and finally, the third one formed by the elevation bands of 4,000 and 4,400 m asl ($J_b = 0.98$) (Fig. 2).

Finally, for the case of the Macizo Colombiano clusters, the statistical GAP method detected a single cluster for both MCW and MCE (*i.e.*, no clusters were detected, ($J_b = 1$ in both cases) (Fig. 2)).

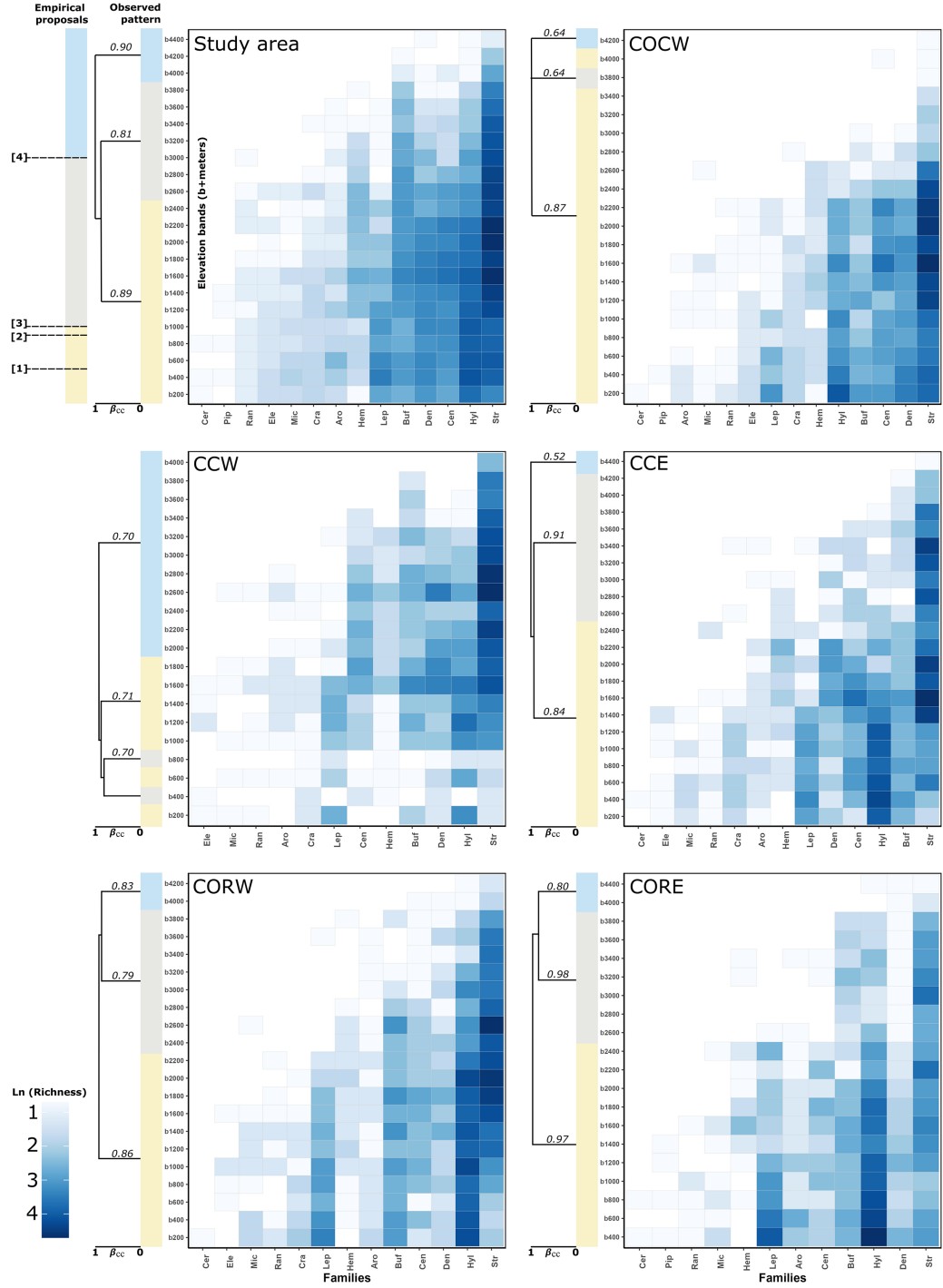

**Figure 2 Heatmaps of anuran families altitudinal distribution and observed clustering (simplified) of elevation bands to study area and each entities with supported dissimilarity group.** In the left-top panel (study area) showed the expected patterns regarding the boundaries taken from empirical proposal: (1) Péfaur & Rivero: 500 m asl, (2) Lynch: 900 m asl, (3) Duellman: 1,000 m asl and (4) Navas—Lynch: 3,000 m asl. The support of the groups in the clusters is given by Jaccard bootstrap (10,000 replicates, numbers on each branch of the clusters). Showed the groups regarding total beta measure (βcc~Jaccard index) (*Carvalho et al., 2013*). A detailed version for the clusters can be found in Fig. S1. Empirical proposals: COCW, Cordillera Occidental western slope; CCW, Cordillera Central western slope; CCE,

**Figure 2** (continued)
Cordillera Central eastern slope; CORW, Cordillera Oriental western slope; CORE, Cordillera Oriental eastern slope; Aro, Aromobatidae; Buf, Bufonidae; Cen, Centrolenidae; Cer, Ceratophryidae; Cra, Craugastoridae; Den, Dendrobatidae; Ele, Eleutherodactylidae; Hem, Hemiphractidae; Hyl, Hylidae; Lep, Leptodactylidae; Mic, Microhylidae; Pip, Pipidae; Ran, Ranidae; Str, Strabomantidae.

Concerning the percentage of species, genera and families spanned by the altitudinal delimitation proposals. We found that, for the study area 38.37% of the species (77.63% of genera, 92.86% of families) presented an altitudinal distribution spanned by Péfaur & Rivero delimitation (500 m asl). 41.19% of the species (75% of genera and 100% of families) presented an altitudinal distribution spanned by the Lynch delimitation (900 m asl). A total of 41.93% of the species (75% of genera and 100% of families) presented an altitudinal distribution spanned by the Duellman delimitation (1,000 m asl). Finally, 18.96% of the species (34.21% of genera and 64.29% of families) presented an altitudinal distribution spanned by the Navas—Lynch delimitation (3,000 m asl) (Table 2, Fig. 2).

Only the family Ceratophryidae had an exclusive altitudinal distribution below 1,000 m asl. The remaining 13 families showed altitudinal distribution (*i.e.* species representation) from 200 m asl to at least 1,400 m asl. No particular species density was observed for any of the families along the elevation bands, nor was there a concordance of the proposed altitudinal delimitations and the observed groupings (Fig. 2).

## DISCUSSION

This study is the first numerical effort, to our knowledge, to evaluate the consistency between the altitudinal distribution of anurans occurring in the Colombian Andes and the delimitation proposals to characterize Andean anurans altitudinally. We present an exhaustive analysis that included 81.33% of the species with records in Colombia. However, we were unable to detect patterns that were consistent with the delimitation proposals tested in this study (*i.e.*, *Duellman, 1979*; *Lynch, 1999*; *Péfaur & Rivero, 2000*; *Navas, 2002*) and our results therefore suggest that these proposals do not reflect the natural differentiation in species composition along the vertical gradient of the mountains, and may indeed be unsuitable for directing conservation decisions for anurans.

*Duellman (1979*, p. 372) proposed to consider as Andean those species of anurans with an elevation distribution above 1,000 m asl, excluding species "primarily from lowlands" and with a peripheral distribution on the Andean slopes. At the study area (Andean level), there was not a single grouping for bands below 1,000 m asl, but these were associated with bands of mid-elevation and the high mountains up to 3,800 m asl (Fig. 2). Moreover, near to 42% of the anuran species richness in the study area had an altitudinal distribution spanned by the Duellman proposal (*i.e.*, its altitudinal distribution has minimum elevation values below 1,000 m asl and maximum elevation values above 1,000 m asl). The results for the other Andean's entities individually are not much different. The number of species whose distribution is spanned by Duellman's proposal varies between 12.78–38.15%

**Table 2 Percentages of species and families of Andean anurans whose altitudinal distribution has minimum values below the altitudinal limits and maximum values above them.**

| Entity | *Péfaur & Rivero (2000)*/ 500 m asl | | | *Lynch (1999)*/900 m asl | | | *Duellman (1979)*/1,000 m asl | | | *Lynch (1999)—Navas (2002)*/ 3,000 m asl | | |
|---|---|---|---|---|---|---|---|---|---|---|---|---|
| | Species (%) | Genus (%) | Families (%) | Species (%) | Genus (%) | Families (%) | Species (%) | Genus (%) | Families (%) | Species (%) | Genus (%) | Families (%) |
| SA | 38.37 | 77.63 | 92.86 | 41.19 | 75 | 100 | 41.93 | 75 | 100 | 18.96 | 34.21 | 64.29 |
| COCW | 36.36 | 48.68 | 85.71 | 34.62 | 44.74 | 85.71 | 35.31 | 44.74 | 85.71 | 1.40 | 2.63 | 14.29 |
| COCE | 21.43 | 48.89 | 92.31 | 24.18 | 51.11 | 76.92 | 21.43 | 37.78 | 76.92 | 1.65 | 4.44 | 15.38 |
| CCW | 17.62 | 43.75 | 91.67 | 25.24 | 54.17 | 100 | 25.71 | 54.17 | 100 | 23.81 | 8.33 | 58.33 |
| CCE | 30.64 | 66.67 | 92.31 | 32.34 | 62.50 | 92.31 | 32.34 | 58.33 | 92.31 | 17.45 | 35.42 | 61.54 |
| CORW | 35.26 | 66.67 | 100 | 38.15 | 66.67 | 100 | 38.15 | 68.89 | 100 | 23.70 | 28.89 | 61.54 |
| CORE | 35.32 | 56.60 | 91.67 | 37.45 | 60.38 | 91.67 | 36.17 | 54.72 | 91.67 | 12.77 | 20.75 | 50 |
| MCW | 14.41 | 23.53 | 58.33 | 15.32 | 20.59 | 58.33 | 15.32 | 20.59 | 58.33 | 10.81 | 11.76 | 25 |
| MCE | 23.31 | 38.46 | 90 | 15.04 | 28.21 | 70 | 12.78 | 28.21 | 70 | 12.78 | 10.26 | 30 |

**Note:**

SA, Study area; COCW, Cordillera Occidental western slope; COCE, Cordillera Occidental eastern slope; CCW, Cordillera Central western slope; CCE, Cordillera Central eastern slope; CORW, Cordillera Oriental western slope; CORE, Cordillera Oriental eastern slope; MCW, Macizo Colombiano western slope; MCE, Macizo Colombiano eastern slope.

among the different entities (Table 2). However, it should be noted that for the entities that presented less than 30% of their species spanned by Duellman's proposal (*i.e.*, COCE, CCW, MCW and MCE) there may be a topographic and sampling effect. In the case of COCE and CCW, these entities are connected by the Cauca River valley (Fig. 1), whose average elevation is 1,000 m asl at its middle region (*Kattan & Franco, 2004*; *Kattan et al., 2004*). Therefore, the percentage of species spanned by Duellman's proposal could be given by the species distributed north and south of the Cauca River valley where the mean elevation is <1,000 m asl (*Kattan & Franco, 2004*; *Kattan et al., 2004*). Although no clusters supported by the statistical GAP method were observed for COCE, the clusters observed in CCW, where lowland bands (<1,000 m asl) were grouped with elevation bands up to 1,800 m asl, suggest that the altitudinal limit proposed by Duellman to separate Andean anurans does not reflect a break in the altitudinal distribution of anuran species in the Cauca River valley. In the case of the Macizo Colombiano, the relatively low percentage of species spanned by Duellman's proposal can be explained by topographic factors (since most of the area of the entities is contained in mountainous regions with elevations >1,000 m asl) and by a relative low number of species records. As can be seen in Fig. 3, the Colombian Massif region (especially in its western and eastern extensions, corresponding to lowlands), presented a low number of records, which can be explained by the historical conflict in this part of the country and which has limited the scientific work in the region (*Reardon, 2018*; *Kolanowska & Szlachetko, 2020*). This lack of records may in turn explain why no clear groupings are observed in the entities of the Macizo Colombiano.

Based on anurans of the ancient genus *Eleutherodactylus* (Currently reassigned between the families Craugastoridae, Eleutherodactylidae and Strabomantidae) of the Colombian Cordillera Occidental, *Lynch (1999*, p. 152) proposed that the anuran distribution may be divided in five different categories, being the lower limit of the Andean species at 900 m asl.

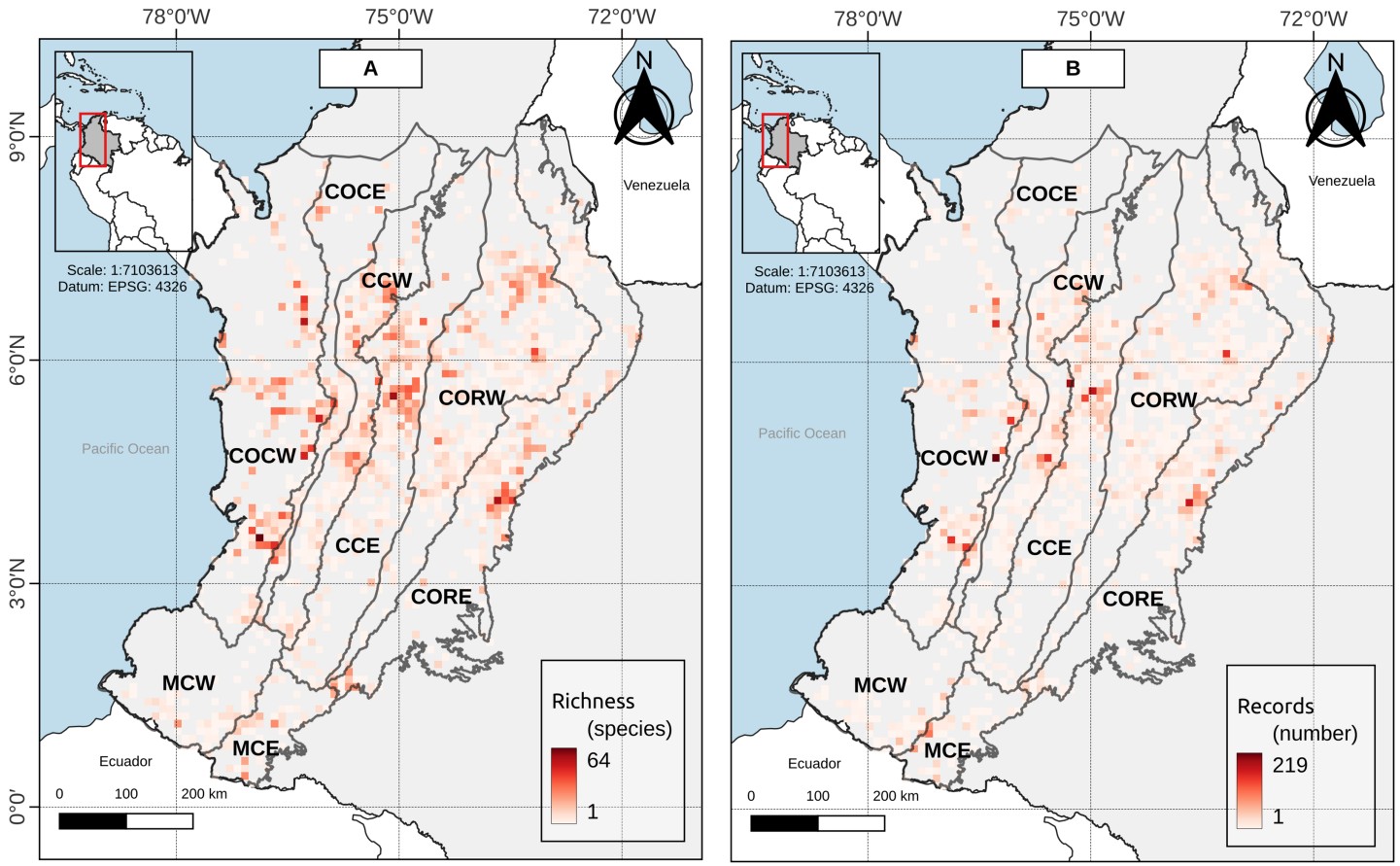

**Figure 3** (A) Species richness and (B) record richness in the study area. Cell size: 0.1° × 0.1° (WGS84—EPSG:4326, 11 km × 11 km approx.).

This division was based on the hypothesis that anurans of this genus (and other groups such as the Centrolenidae frogs) were distributed across equivalent bands on the Andean slopes. Our results suggest that elevation bands lower than 900 m do not constitute a clearly distinct group of species, neither at the level of the study region, nor for any of the entities individually, so we could suggest that Lynch's altitudinal delimitation to separate lowland frogs and Andean frogs does not reflect a break in the altitudinal composition of anurans in the Colombian Andes. In terms of the percentage of species whose altitudinal distribution is spanned by Lynch's proposal, the panorama is very similar to that found with Duellman's proposal, since only COCE, CCW, MCW and MCE presented less than 25% of their species spanned by Lynch's proposal, and the reasons for this discrepancy with the other entities can also be explained by topographic reasons and a relative low number of species records (see discussion for *Duellman (1979)* delimitation).

*Péfaur & Rivero (2000*, p. 45) proposed that species with an elevation distribution above 500 m asl could be considered Andean, and those with a distribution below this elevation could be considered as foothill species. For the Colombian Andes region, we found that elevation bands lower than 500 m asl do not constitute a differentiated group, but were associated with elevational bands of medium and higher elevations (see above). Similar to

the proposals for a boundary between lowland anurans and Andean anurans by *Duellman (1979)* and *Lynch (1999)*, the percentage of species whose altitudinal distribution is spanned by the proposal of Péfaur & Rivero, varies between 14.41–36.36%. Being the entities with a percentage lower than 30% those belonging to the Cauca River valley (COCE and CCW) and those of the Macizo Colombiano (MCW and MCE), whose topography and a relative low number of species records (Figs. 1 and 3) can explain a lower number of spanned species. However, in none of the cases, neither for the study area nor for the entities was observed that the elevation bands below 500 m asl formed a distinct group, so we can suggest that the proposed delimitation of Péfaur & Rivero does not reflect a break in the altitudinal composition of anurans in the Colombian Andes.

In the case of the upper delimitation of the Andes (*Lynch, 1999*; *Navas, 2002*), we found that, for the study area, there is a differentiation of two groups above 2,600 m asl, one grouping the elevation bands between 2,600–3,800 m asl and the second grouping the bands between 4,000–4,400 m asl. For all entities where clusters could be differentiated, three groupings were observed, however their boundaries and composition varied between entities, and in no case did the elevation bands >3,000 m asl form a distinct group from the others. Therefore, an upper limit of 3,000 m asl for the differentiation of Andean anurans and high mountain anurans does not seem to reflect a break in the altitudinal distribution of anurans distributed in the Colombian Andes.

The results obtained in the heat-maps (Fig. 2), showed that all of the anuran families in the study area are distributed across the entire elevation gradient (except for Ceratophryidae). Variation in the elevation distribution of species was more or less constant among families present in the region (Table 2). These patterns could be explained by the age of the Andean mountain range, which began to emerge during the early Miocene (~23 MA) and finally consolidated in Colombia in the Pliocene and Plestoic (~2.5-2 MA), when the mountains reached their current or slightly higher elevations (*Hernández-Camacho, 1992*; *Guariguata & Kattán, 2002*). On the other hand, the most recent anuran families diverged during the early Cenozoic in the middle of the Paleogene up to ~50 MA (*Vitt & Caldwell, 2014*) and the uprising of the Colombian Andes is therefore a relatively recent phenomenon compared to the history of anuran diversification, which could be the reason why most of the Colombian anuran families are distributed along the elevation gradient of the Andes. Moreover, authors such as *Navas (2002)* and *Acosta-Galvis (2015)* have suggested that the success of anurans in the Andes mountain range reflects physiological plasticity within species and that it seems to be a characteristic of this group across different families. Thus, the plasticity of anuran species could be another explanation for why most of the families are represented along the elevation gradients of the different Andean slopes.

At the study area level, we found three elevation groups: 200–2,400 m asl, 2,600–3,800 m asl and 4,000–4,400 m asl. These patterns are well supported ($J_b > 0.8$) and differ from the evaluated delimitation proposals. However, this clustering was not consistent across the different entities. Although three elevation groups were formed in all entities where clustering was detected, we do not have a sufficiently consistent pattern among entities to suggest a discrete, altitudinal distribution-based division that can be

applied to all entities and all anurans distributed in the Colombian Andes. We are not aware of efforts with other vertebrate groups to define such compositional breaks (*i.e.*, altitudinal boundaries) in the Andes region. However, given that altitudinal classifications have been and continue to be used for anurans in the Andes (*e.g.*, *Bernal & Lynch, 2008*; *Anderson et al., 2011*; *Meza-Joya & Torres, 2016*; *Armesto & Señaris, 2017*), we may suggest the use of biogeographic proposals such as those of *Hernández-Camacho (1992)* or *Morrone (2014)*, to define "Andean regions" through the union of multiple biogeographic provinces. The elevation groups found for the study region could also be used. However, it is necessary to increase the sampling effort (number of records) in COCW and the Macizo Colombiano, where no clusters were detected, to see how these entities are grouped and how their groupings influence the general pattern of clustering in the region.

## CONCLUSIONS

In general, it was observed that the different empirical delimitations proposed for anurans of the Andes did not coincide with the elevational band groups detected in this study. It is important to highlight that inter-Andean valleys do not behave as distinct entities from the mountain ranges, and should therefore not be excluded in future studies of species lists or conservation. Considering that decision makers, at best, rely on information produced in academia, it is our responsibility to provide the least biased possible information. This is why we emphasize that the use of arbitrary criteria in our methodologies can lead us to misinterpret natural patterns. We propose three ideas for consideration when conducting studies with anurans in the Colombian Andes: (i) do not exclude inter-Andean valleys, (ii) the distribution of anurans in Colombian Andes should be considered across their full distributional range, and avoid using any delimitation that arbitrarily span the altitudinal distribution of species, and (iii) although there appears to be an altitudinal clustering for anurans in the study region, information is still lacking in some areas and there is not yet a generalizable clustering pattern. Thus additional criteria such as natural history or phylogenetic relationships should be considered in order to define the limits of the study.

## ACKNOWLEDGEMENTS

We thank Eduardo Pineda-Arredondo, Larry Jiménez and Jenifer Girón for reviewing the first version of the manuscript and for their valuable comments. We also thank the three anonymous reviewers whose comments and suggestions substantially contributed to the improvement of this work from the first version submitted.

### Funding

This work derives from JMHL's undergraduate dissertation at the Universidad de Caldas (Colombia), funded by CONACYT project support funds (Chair project No. 673) and SNI's Dr Cultid-Medina and for Facultad de Ciencias Exactas y Naturales, Universidad de Caldas (Colombia). The publication fees of this article were covered by the Vice-Rector for Research and Postgraduate Studies of the Universidad de Caldas (notice N. 01 of April 11,

2023) as academic support for the MC Viviana Ramírez and for operating funds of the Biodiversity in Neotropical Landscapes Lab (BNP-Lab) directed by Dr Cultid-Medina. The funders had no role in study design, data collection and analysis, decision to publish, or preparation of the manuscript.

### Grant Disclosures

The following grant information was disclosed by the authors:
CONACYT: (Chair project No. 673).
Universidad de Caldas (Colombia).
Biodiversity in Neotropical Landscapes Lab (BNP-Lab).

### Competing Interests

The authors declare that they have no competing interests.

### Author Contributions

- Jorge Mario Herrera-Lopera conceived and designed the experiments, performed the experiments, analyzed the data, prepared figures and/or tables, authored or reviewed drafts of the article, and approved the final draft.
- Viviana Andrea Ramírez Castaño performed the experiments, prepared figures and/or tables, authored or reviewed drafts of the article, and approved the final draft.
- Carlos A. Cultid-Medina conceived and designed the experiments, analyzed the data, prepared figures and/or tables, authored or reviewed drafts of the article, and approved the final draft.

### Data Availability

The raw data and code are available in the Supplemental Files.

### Supplemental Information

Supplemental information for this article can be found online at http://dx.doi.org/10.7717/peerj.15217#supplemental-information.

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
