# Peer review of "What are the Andean Colombian anurans? Empirical regionalization proposals vs. observed patterns of compositional dissimilarity"

_PeerJ, doi:10.7717/peerj.15217_

## Round 0.1 · original submission · Major Revisions

After receiving the comments of three reviewers who sent important and detailed comments on the paper "What are the Andean Colombian anurans? Empirical regionalization proposals vs. observed patterns of compositional dissimilarity" I consider that a number of Major Revisions are required. It is important that the observations of the three reviewers are made, mainly to modify the discussion to only discuss the results that were obtained since currently there are many conclusions that are not supported by the statistical analysis.

·

Basic reporting

This study tries to condense four different altitudinal criteria that have been established to define Andean anurans, when it is in fact unclear whether it is appropriate to use such a generalized definition of Andean anurans.
The Introduction and background are clear in demonstrating the context and the problematic of this denomination for Andes anurans.
The Literature is well referenced and relevant.
The structure of this manuscript is professional, in accordance to PeerJ standards and discipline norm.
In general, the three figures of this study are relevant, but require further editing to improve their quality increasing the resolution, the labeling and description.
The submission is self-contained, because represent an appropriate unit of publication, and it include all results relevant to the hypothesis.

Experimental design

This study is an original primary research within Aims and Scope of the PeerJ.
Research question is well defined and relevant. It is stated how this research fills an identified knowledge gap for Andes anurans, and statements are made as to how the study contributes to filling that gap.
This investigation has been conducted rigorously and to a high technical standard, in conformity with the prevailing ethical standards in the field.
The Methods describe the study area, the collection of records for Colombian Andean anurans and the data analysis. All are described with sufficient information to be reproducible by another investigator.

Validity of the findings

All underlying data have been provided; they are robust because a large sample size (species records) is included, statistically sound because are based in Cluster analysis, and controlled under an altitudinal criterion.
Conclusions are well stated and connected to the original question investigated, and limited to supporting results.

Additional comments

In the introduction is cler that there are no studies with amphibians that delimit them altitude wise, but it is necessary to define if these types of studies do not exist for other groups of organisms in the Andes either.

I consider that this study can increase its contribution to the resolution of the stated problem, if it were possible to associate the changes in the altitudinal diversity of amphibians with the altitudinal changes of the ecosystems in the region

The division of the Colombian Andes into tres parts (Occidental Cordillera, Central Cordillera and Oriental Cordillera) is unclear for the reader who is unaware of the local regionalization of Colombia, and this regionalization should be indicated within Fig. 1 to clearly understand the configuration of the spatial division.

In this study, occurrence records were obtained from literature records and local databases, however, I consider that they should include international databases such as, Global Biodiversity Information Facility (GBIF; https://www.gbif.org), HERPNET (http://www.herpnet.net/), the Vertebrate Network (VerNet, http://vertnet.org/index.html).

I consider that in the discussion it could be considered if the pattern of altitudinal distribution of Andean anurans is coincident with some other taxa

The text inside figure 2 is difficult to read, the font size should be increased.

The symbology of the figures is not defined within the captions of figures.

Reviewer 2 ·

Basic reporting

The English language use is clear throughout, and carefully written. However, with respect to the other conditions, the manuscript does not meet the expected requirements: the cited literature is outdated and insufficient, context on the research question is lacking --the discussion is centered on criticism instead of providing arguments/evidence to allow the reader to compare, rate and decide on a limited number of proposals, which are regarded as erroneous. And the announced improvements to be made in order to have a robust delimitation between "Andean" and "lowland" amphibians fade across the little informative tables and the main figure (Figure 3), which is supposed to support the findings but it is too difficult to follow. There are particular comments on other tables and figures, which should be considered in order to make them clearer.

Results are supported, but of scarce novelty and the most novel subject of discussion (the inclusion of the inter-Andean valleys in the Andean regionalization) has not been clearly addressed.

Experimental design

The scope of the paper complies to the aims and scope of the journal, but, from my point of view, the relevant question presented in the tittle is not answered. The evidence provided and analyzed is just a part of the evidence needed to start solving the question of which amphibian species in Colombia should be called Andean (based solely in altitudinal ranges). Besides, even the relevance of the question is being discussed in recent literature (since the delimitation of paramos, for example), and nothing about that was mentioned in the text.

The sources of distributional information used are very, very, limited, especially considering that there exists a Global Biodiversity Information Facility, and a Biodiversity Information System of Colombia, which make publicly available the basic information that the authors want and need to use.

The provision of an R script linked to a database should be helpful for readers and it should guarantee replicability. However, the criteria used to build the amphibian database linked to the program (594 sp) are not clear. Why did the authors include only 593 species of the nearly 900 known to date for Colombia?

Validity of the findings

The findings are the logic consequence of the data and methods used, but they do not provide clarity on how an "Andean" amphibian might be recognized in Colombia. Considering that the usage of the term is tricky, more extensive support should be provided to discuss it, or to propose new insights on the subject.

Reviewer 3 ·

Basic reporting

This study evaluates the consistency of four empirical regionalization proposals for the anurans of the Andean region of Colombia using presence records of 593 species. The authors used a simple, yet effective approach (cluster analyses), to contrast their hypothesis. The work is a novel and original contribution to the knowledge about this biological group. The results include clear evidence to support a change in the current biogeographical categorization of the Andean anurans. Nonetheless, several issues should be amended before the publication of this work.
The article is well written in general, nonetheless, I think some ideas should be clarified. For example, in lines 122 to 132, the purpose of this paragraph is unclear because the idea of the topographic/climatic complexity at the local scale is not included in the objective or the hypothesis.
I think that the caption of figure 1 should be changed, I got confused when reading about four divisions (lines 168 to 181), but in the figure, it appears 8 acronyms (entities). This confusion is clarified through the reading of the article, nonetheless, it should be clearly stated in the figure caption. It should be considered to change the text in lines 168 to 181 because the division of the Colombian Andes into four parts may be useless, as it is not used for analysis or discussion purposes. In figure 3 it is hard to distinguish between the geographical thresholds, particularly between Péfaur and Rivero (2000), and Navas (2002). I recommend changing the dashed line style in one of these thresholds.
Abstract. I think that the description of the study context is too long. I recommended synthesizing this section. The objective of the study is somehow ambiguous in this section, I suggest a more direct and concise redaction. Finally, the conclusion is unclear, please state clearly what is the main conclusion of your work

Experimental design

The use of presence records of the anuran species is the basis of the analysis, and it is clearly explained. The number of records per species is a very important issue of the analysis because it is directly related to the known distribution of a species. This information is missed, and it must be added to clearly determine the validity of the analysis. I think that scarcely represented species in the distribution records should be excluded from the analysis or it must be clearly explained and justified why those species (if present) are retained.
I think that the research question of this work is clear and relevant. The authors have used a simple approach to analyze the research question (cluster analysis). I think that a more detailed analysis of the beta diversity (e. g. beta partitioning), could be more informative, nonetheless, the question at hand is solved through the approach used; thus, a more detailed analysis is not mandatory.

Validity of the findings

The discussion section is too descriptive (It seems somehow a detailed description of the results), I think it should be restructured to include an organized contrast with the current knowledge and a meaningful implication of the results.
I suggest modifying the conclusion redaction to make it more concise.

---

## Round 0.2 · Major Revisions

Dear Authors,

The manuscript has been improved a lot with the previous observations, however one reviewer still has some important observations about the analyses carried out, mainly they comment that the number of species with few occurrence records may be affecting the analyzes carried out, as well as some details of wording and order of the ideas in the results section, for this reason, major corrections are necessary in order to be accepted the manuscript, however, the manuscript has been greatly improved since the first version.

Sincerely,

Armando Sunny.

Reviewer 3 ·

Basic reporting

no comment

Experimental design

No comment

Validity of the findings

My main concern is that you mention that the sampling effort (I think that you should be referring to records of occurrence instead), may be affecting the observed altitudinal distribution pattern of amphibians (line 492, 562, 584, and 635). I agree with such idea, and as mentioned in the previous review, I think that species with few occurrence records may be affecting your interpretations; thus, an analysis of the number of records per species is necessary in order to determine the reliability of the interpretations. I know that you have provided a table with all the records per species in the supplementary material; nonetheless I think that an explicit analysis of this issue must be included in the methods and results section. Otherwise, you must stablish and mention some criteria to include and/or exclude species from your analysis. For example, date of the record, minimum number of records per species, etc. I strongly suggest avoiding using species with too few records of occurrence due to its geographic distribution is underestimated.

Additional comments

I think that the authors had improve the manuscript using the comments on the first review. Nonetheless, I still have some issues that must be attended prior publication on this work.
1. My main concern is that you mention that the sampling effort (I think that you should be referring to records of occurrence instead), may be affecting the observed altitudinal distribution pattern of amphibians (line 492, 562, 584, and 635). I agree with such idea, and as mentioned in the previous review, I think that species with few occurrence records may be affecting your interpretations; thus, an analysis of the number of records per species is necessary in order to determine the reliability of the interpretations. I know that you have provided a table with all the records per species in the supplementary material; nonetheless I think that an explicit analysis of this issue must be included in the methods and results section. Otherwise, you must stablish and mention some criteria to include and/or exclude species from your analysis. For example, date of the record, minimum number of records per species, etc. I strongly suggest avoiding using species with too few records of occurrence due to its geographic distribution is underestimated.
2. There is inconsistencies on the abbreviations of the entities, e.g., in line 334 you use the abbreviation CORW for the easter slope of the Cordillera Central, whilst in the study area section the abbreviation for this entity is CCE. Such inconsistencies produce great confusion.
3. According to the objective and the introduction, the evaluation of the empirical delimitation proposals for amphibians is the main purpose of the study (is also one of the first ideas in the discussion section), but in the results section this issue is presented after the analysis of the elevation bands. I suggest changing the order in which these results are presented to maintain the same order as in the other sections.
4. In lines 422-430, you mention that figure 3 may be used to illustrate the percentage of species, genera and families crossed by the altitudinal delimitation proposals. I found hard to see such relation in figure 3, you may clarify such relation.

---

## Round 0.3 · accepted · Accept

The new version of this manuscript shows an improvement in all the points indicated in the previous revisions, for this reason the article is now ready to be accepted.

·

Basic reporting

The new version of this study shows an improvement in all the points indicated in the previous review. This study tries to condense four different altitudinal criteria that have been established to define Andean anurans. In this new version the authors redefined the appropriate generalized definition of Andean anurans.
The Introduction and background are clear in demonstrating the context and the problematic of this denomination for Andes anurans.
The Literature is well referenced and relevant.
The structure of this manuscript is professional, in accordance to PeerJ standards and discipline norm.
Tables and figures were improved. The three figures of this study are relevant, and in this new version its edition and quality are increased. Figures 2 and 3 were replaced by new figures that better highlight the patterns found in this study, improved the font size and symbology.
The submission is self-contained, because represent an appropriate unit of publication, and it include all results relevant to the hypothesis.

Experimental design

This study is an original primary research within Aims and Scope of the PeerJ.
Research question is well defined and relevant. It is stated how this research fills an identified knowledge gap for Andes anurans, and statements are made as to how the study contributes to filling that gap.
This investigation has been conducted rigorously and to a high technical standard, in conformity with the prevailing ethical standards in the field.
The Methods describe the study area, the collection of records for Colombian Andean anurans and the data analysis. All are described with sufficient information to be reproducible by another investigator.

Validity of the findings

All underlying data have been provided; they are robust because a large sample size (species records) is included, statistically sound because are based in Cluster analysis, and controlled under an altitudinal criterion.
Conclusions are well stated and connected to the original question investigated, and limited to supporting results.

Additional comments

In the introduction is cler that there are no studies with amphibians that delimit them altitude wise, in this new version it is already defined that these types of studies do not exist for other groups of organisms in the Andes either.
The division of the Colombian Andes into tres parts (Occidental Cordillera, Central Cordillera and Oriental Cordillera) is now better defined, and this regionalization is now included within Fig. 1.
In this study, occurrence records were obtained from literature records, local and now global databases as GBIF.
The discussion is now more focused and cautious, highlighting the methodological criteria used within the study to consider regional altitudinal differences. The conclusions of the study are also now more cautious and limited to the results found.